# Cell-Penetrating Peptides for Use in Development of Transgenic Plants

**DOI:** 10.3390/molecules28083367

**Published:** 2023-04-11

**Authors:** Betty Revon Liu, Chi-Wei Chen, Yue-Wern Huang, Han-Jung Lee

**Affiliations:** 1Department of Laboratory Medicine and Biotechnology, College of Medicine, Tzu Chi University, Hualien 970374, Taiwan; 2Department of Life Science, College of Science and Engineering, National Dong Hwa University, Hualien 974301, Taiwan; danielcwchen@gms.ndhu.edu.tw; 3Department of Biological Sciences, College of Arts, Sciences, and Education, Missouri University of Science and Technology, Rolla, MO 65409, USA; huangy@mst.edu; 4Department of Natural Resources and Environmental Studies, College of Environmental Studies and Oceanography, National Dong Hwa University, Hualien 974301, Taiwan

**Keywords:** cell-penetrating peptides, transgenic plants, gene delivery, direct membrane translocation

## Abstract

Genetically modified plants and crops can contribute to remarkable increase in global food supply, with improved yield and resistance to plant diseases or insect pests. The development of biotechnology introducing exogenous nucleic acids in transgenic plants is important for plant health management. Different genetic engineering methods for DNA delivery, such as biolistic methods, *Agrobacterium tumefaciens*-mediated transformation, and other physicochemical methods have been developed to improve translocation across the plasma membrane and cell wall in plants. Recently, the peptide-based gene delivery system, mediated by cell-penetrating peptides (CPPs), has been regarded as a promising non-viral tool for efficient and stable gene transfection into both animal and plant cells. CPPs are short peptides with diverse sequences and functionalities, capable of agitating plasma membrane and entering cells. Here, we highlight recent research and ideas on diverse types of CPPs, which have been applied in DNA delivery in plants. Various basic, amphipathic, cyclic, and branched CPPs were designed, and modifications of functional groups were performed to enhance DNA interaction and stabilization in transgenesis. CPPs were able to carry cargoes in either a covalent or noncovalent manner and to internalize CPP/cargo complexes into cells by either direct membrane translocation or endocytosis. Importantly, subcellular targets of CPP-mediated nucleic acid delivery were reviewed. CPPs offer transfection strategies and influence transgene expression at subcellular localizations, such as in plastids, mitochondria, and the nucleus. In summary, the technology of CPP-mediated gene delivery provides a potent and useful tool to genetically modified plants and crops of the future.

## 1. Introduction

The current trends in crop yield fall short of meeting the demand, as the global requirement for food is projected to double in the next 30 years [1]. Modern agriculture is facing major global challenges, such as loss of biodiversity, chemical contamination of soils, plant pests, and diseases [2], all of which can directly affect plant health and productivity. Genetically modified plants and crops provide one of the solutions to increase global food production with improved gains in yield and resistance to plant diseases or insect pests. Several successful cases of genetically modified plants have conferred phytoprotection against insects, pests, and pathogens, such as overexpression of proteinase inhibitor genes from legumes [3], recombinant Bt toxic proteins from soil bacteria *Bacillus thuringiensis* [4], α-amylase inhibitors, and plant lectins [5]. In spite of these successful examples, there is a need to develop alternative strategies of phytoprotection.

Transgenic plants are defined as plants containing gene modifications and expressing recombinant proteins or products from foreign genes [6]. The success of transgenic plants depend on favorable methods of gene delivery. The first transgenic plant was reported in 1983 when an antibiotic-resistant Ti plasmid was delivered into tobacco, mediated by *A. tumefaciens* [7]. Subsequently, tremendous gene delivery strategies, such as particle bombardment (biolistics), were applied in plants. Flourishing developments of biotechnology in exogenous nucleic acid delivery have brought a great improvement in transgenic plants [8,9].

Genetic transformation methods in plants were generally divided into two types: direct and indirect gene delivery methods [9]. DNAs or RNAs can be introduced into plants either directly or packaged by specific viruses or bacteria, then transferred into plants via an indirect method [9]. Additional gene delivery classifications include physical, chemical, and biological methods (Figure 1) [10]. Physical methods, such as microinjection, biolistics, electroporation, silicon carbide fibers, laser-mediated DNA delivery, sonoporation, hydrodynamic force, etc., facilitate nucleic acids to penetrate cell membrane directly [11]. The advantages of physical methods for gene delivery are plant- and genome-type independence. Large- or small-sized plasmid DNAs can be delivered by these methods, while DNA transformations in some recalcitrant plants, such as cereals and legumes, are wildly applied [12]. However, the criticisms of physical methods are irreversible tissue damage and integration of genes into host genomes [10]. In microinjection and electroporation methods, plant sample preparations become protoplasts or a single cell, and this complicated procedure makes the drawback for transgenic plant applications [10].

Polyethylene glycol (PEG)-, diethylaminoethanol (DEAE)-dextran-, calcium phosphate-, dendrimer-, and liposome-mediated gene transfers were categorized as chemical methods [11,13]. Hygromycin resistance gene was introduced into the protoplasts of rice, and these transgenic rice plants generated viable seeds [14]. The benefits of liposome-mediated nucleic acids delivery include DNA protection from nuclease digestion, as well as suitability for multiple types of plant cells, such as protoplasts and plasmodesmata [15]. Calcium phosphate is a cheap and easily handled DNA delivery method that possesses economic benefits. DNAs are able to interact with positively charged calcium ions via electrostatics, form precipitates, and enter cells by endocytosis [11]. Endocytosis is the major route for DNA delivery in chemical method-mediated plant transformation [16]. However, nucleic acids, which might be digested in lysosomes, and the limitation on plasmid DNA size reduce the potential of application in transgenic plants [16,17].

*A. tumefaciens* is the earliest and most common biological method for DNA delivery in transgenic plants (Figure 1) [7,12]. This bacterium has the ability to deliver genes into the host genome, leading to the establishment of *Agrobacterium*-mediated transformation as one of the standard protocols in transgenic plants [8,12]. However, there are some disadvantages in *Agrobacterium*-mediated transformations [18,19]. First of all, low transformation yield was reported in certain plant species, which might be due to the specificity of hosts [19]. This reduces the enthusiasm in transgenic engineering in plants. Additionally, it took long tissue culture periods to recover from transformation, and low yield of stably transformed plants was observed in *Agrobacterium*-mediated transformation [8,18,19]. Due to these limitations, virus-based vectors become an alternative (Figure 1). Protein expressions by virus-based infected genes are able to be produced more quickly and have a greater yield [20]. However, safety concerns and continuous re-infection are the drawbacks for this method [7]. Recently, non-viral peptide-based gene delivery systems have become popular in transgenic engineering. However, not all peptides possess the same abilities for gene delivery. The membrane-active peptides, cell-penetrating peptides (CPPs), were reported as a good tool for nucleic acid transfection [21,22]. We devote the sections below to introduce CPPs and their roles in transgenic plants in detail.

## 2. Cell-Penetrating Peptides (CPPs)

CPPs are short and membrane-active peptides [21]. In general, macromolecules, such as DNAs, RNAs, and proteins, are impermeable to cell membranes. The cell membrane is a natural barrier to prevent harmful exogenous or pathological molecules from entering the cell freely, and to maintain the osmotic balance within the cell. Some functional proteins are able to enter cells via specific receptors or channels, while nucleic acids alone are generally not [23]. Not only can CPPs enter cells by themselves, but also deliver various cargoes, including nucleic acids, into living cells [24]. However, nucleic acids are not the only macromolecules that CPPs are able to deliver. CPPs also can serve as a Trojan horse, while peptides/proteins [25], nanoparticles [26], pharmaceutical molecules, and small drugs [24,27,28] play the role of Achilles. The selectivity and efficiency of drug/molecules delivery are significantly improved when CPPs cooperate with liposomes/micelles [29,30]. Most CPPs have been shown to be nontoxic, and do not interfere with functionality of the delivered biomacromolecule [31].

There are various interactions between CPPs and their cargoes. CPPs and cargoes can form complexes with covalent bonds [32], noncovalent interactions [21], and covalent- and noncovalent-synchronous linkages [33]. The potential of bio-membrane penetration in CPPs is amazing. Until now, studies indicated that CPPs were able to penetrate different targets, including mammalian cells [21], plant cells/tissues [32,33,34,35,36,37,38,39], rodent skin and intestinal mucosa [25], prokaryotes [40,41], fungi [42], insect cells [43], paramecia [44], and rotifers which were individual organisms containing thick cuticles [45]. Recently, different modifications on CPPs, such as D-form amino acid applications, branches on backbone sequences, cyclic structures alterations, and non-standard amino acid substitutions were designed to increase their internalization efficiency and stability [21,46,47,48]. Highly cellular penetration efficiencies and non-cytotoxic properties make CPPs an ideal delivery system for therapeutic drugs, gene therapies, and transgenic plants [22,25,49,50,51].

### 2.1. Categories of Cell-Penetrating Peptides

Since the first CPP, trans-activator of transcription (Tat) protein of the human immunodeficiency virus type 1 (HIV-1), was identified, 1855 CPP entries with sequence information have been deposited and annotated in a database repository named the CPPsite 2.0 [52]. In this database, CPPs were categorized according to various characterizations, such as peptide lengths, chemical and physical properties, and structures [52]. Additionally, it offered further predicting of CPPs. Taking peptide length as an example, there were 60 CPPs containing five or less residues in this CPPsite 2.0 online database. L5a (Table 1), the peptide composed of only five amino acid residues, was one of them [53]. Some novel peptides or protein segments were also predicted as CPPs by the machine learning web-server of KELM-CPPpred [54]. Classification of CPPs can help identify a suitable CPP for a specific purpose. Three types of CPPs were categorized based upon their chemical and physical properties [55].

#### 2.1.1. Cationic Type

The Tat protein transduction domain (PTD) is the most well-known representative of a cationic CPP. When Tat protein, consisting of 86 amino acid residues, was first discovered by two independent groups in 1988 [84,85], researchers found that the Tat PTD (amino acid residues 48–60) was composed of eight cationic amino acids [56]. These cationic amino acids were regarded as the key for cellular translocation. Replacing the positively charged residues with other amino acids decreases cell entry efficiency [59,86]. Later, many derivatives from the Tat PTD were made. Among them, poly-arginines, such as octa-arginine (R8) and nona-arginine (R9), display more amazing cellular uptake efficiency [21,59,86]. Besides the Tat PTD and poly-arginines, penetratin, the third helix of the homeodomain of *Antennapedia* protein (pAntp), diatos peptide vector 1047 (DPV1047), PTD-5, and poly-lysines were among the cationic CPPs [55]. Studies indicated that peptides containing much richer arginines correspond with higher cellular penetration efficiencies than the peptides with lysine-rich residues [55,59]. Peptide structures were studied and the guanidinium groups on arginines were considered as the crucial factor for membrane disturbance and peptide entry [55].

#### 2.1.2. Amphipathic Type

Amphipathic CPPs are peptides that contain both polar and nonpolar regions of amino acids. VP22, KALA, GALA, model amphipathic peptide (MAP), Pep-1, VE-cadherin-derived peptide (pVEC), alternative reading frame (ARF), and MPG are the examples of amphipathic CPPs [29,55]. They may contain valine, leucine, isoleucine, and alanine to form the hydrophobic portions on their secondary structures, and their hydrophilic portions are composed of charged residues like lysine, arginine, and histidine [55]. Positively charged portions on amphipathic CPPs increase the solubility and offer high affinity to cell membrane with electrostatic interaction, while the hydrophobic portions are able to fuse with lipids easily and allow peptides to enter cells by membrane-inverted fusions [54]. Due to these two portions, α-helix (hydrophilic propensities) and β-sheet (hydrophobic propensities) are easily observed in amphipathic CPPs [55].

#### 2.1.3. Hydrophobic Type

The number of hydrophobic CPPs is fewer compared to two other types of CPPs. In this hydrophobic family, peptides contain only non-polar residues and limited net charges [55]. C105Y, PFVYLI (derived from the c-term of C105Y), and Pep-7 were representatives [55]. However, some studies suggested that MAP, which contained both lysine and non-polar residues and was categorized in the amphipathic group, should belong to the hydrophobic CPP [22,87]. The reason for this sorting was based upon the lack of cell selectivity. Hydrophobic CPPs usually display a wide penetrating ability in various tissue types by membrane disorganization [87]. As shown, there are many pros of hydrophobic CPPs and MAP containing these properties. Unfortunately, the cons come from the same properties. Their entry mechanisms lead to higher hemolysis and membrane pore formations, resulting in adverse side effects and limitations in applications [87].

### 2.2. Mechanisms of Cellular Internalization of Cell-Penetrating Peptide/Cargo Complexes

Although mechanisms of cellular internalization of CPPs have been continuously studied, the discrepancies in understanding their entry routes remain considerable. Many factors, such as primary sequences of CPPs, modifications on peptides, types of cargoes, concentrations of CPPs, linkages between CPPs and cargoes, as well as entry targets of cell lines, can influence penetrating mechanisms [21,22]. Up to date, energy-dependent endocytic pathway and energy-independent direct membrane translocation seem to be the major uptake routes [22]. As shown in Figure 2, in direct membrane translocation, low temperature and endocytic inhibitors cannot stop the entry of CPPs or CPP/cargo complexes [66]. The process of this mechanism starts with attaching CPPs or CPP/cargo complexes to lipid membrane utilizing electrostatics or hydrophobic interactions, followed by membrane destabilization [55,87]. Direct membrane translocation can be further divided into three different pathways: inverted micelles, carpet model, and pore formation [57,88]. For the inverted micelles pathway, CPPs or CPP/cargo complexes attach to membrane surface, followed by invagination of lipid bilayers [60]. Liposome-like structure will form later to package the CPPs or CPP/cargo complexes, and phospholipid inversion flips CPPs or CPP/cargo complexes to enter the cytoplasm (Figure 2). The carpet model is a concentration-dependent model, which is usually applied in α-helical cationic CPPs [89]. CPPs remain parallel to the surface without insertion at low concentrations, while changing the membrane fluidity by redirection of phospholipids and making micelles and pores at high concentrations. Pore formation, also called the Barrel-Stave model, occurs in CPPs with strong positive charge [88], such as HR9 (charge: +14 calculated by CellPPD website [90]) or amphipathic CPPs with an α-helical structure, such as mastoparan [91]. CPPs or CPP/cargo complexes enter cells by direct membrane translocation to avoid the trap from lysosomes, but this action causes highly disturbed membranes, leading to a high risk of cellular injury. Energy-dependent endocytosis is also reported in some CPPs or CPP/cargo complexes [21,22,55].

Endocytosis is a natural process involving at least four subtypes: clathrin-mediated endocytosis, caveolae-mediated endocytosis, clathrin- and caveolae-independent endocytosis, and macropinocytosis [21,38]. Both clathrin-mediated and caveolae-mediated endocytic pathways involve receptor absorptions, microtubule and actin rearrangements, while macropinocytosis (Figure 3), an atypical endocytic route, only involves actin rearrangements [21,69,92,93]. It is hard to conclude whether the types of CPPs or CPP/cargo complexes highly correspond to the types of endocytosis. Studies indicated that CPPs or CPP/cargo complexes enter cells primarily through endocytosis [22,55]; however, SR9 and PR9 enter cells by endocytosis as well [21,69]. Furthermore, SR9 carrying nanoparticles enter cells by multiple endocytic pathways [26].

The relationship between the types of CPPs and their entry mechanisms is unclear. Most studies indicated that cationic CPPs interact with cell membranes by electrostatics and cause disruption of lipid bilayer [94]. This process either thins the cell membrane, according to the membrane thinning model, or makes some pores in cellular membranes, according to the pore formation model [94]. This led Xie et al. to propose that most cationic CPPs enter cells by direct membrane translocation [88]. Pore formations easily perturb the membrane stability, which raises the concern of cytotoxicity. However, not all cationic CPPs enter cells by direct membrane translocation. For instance, SR9 and PR9, two cationic CPPs, use endocytosis for cellular entry (Table 1).

Amphipathic CPPs possess both hydrophobic and hydrophilic properties, which are considered to be a key factor for intracellular internalization [95]. As shown in Table 1, most amphipathic CPPs, such as VP22, KALA, and GALA, follow the endocytic pathway. However, there are some amphipathic CPPs, like MPG and Pep-1, which use direct membrane translocation. Derakhshankhah and Jafari proposed that the hydrophobic portions of amphipathic CPPs insert into cellular membranes by hydrophobic force and polymerization, while the hydrophilic portions of amphipathic CPPs form pores in cellular membranes [94]. However, this hypothesis cannot fully explain the complexity of cellular internalization as other factors may also be involved in cellular entry. Target cells/tissues/species are also determinants in cellular internalization. SR9/quantum dot complexes entered A549 cells and prokaryotes by using multiple pathways [26] and macropinocytosis [68], respectively. Kauffman et al. indicated that cationic CPPs enter cells by endocytosis at low concentrations (<10 µM), and switch to direct membrane translocation at high concentrations [96]. The authors also suggested that high concentrations of cationic CPPs might have caused membrane disruption and possibly increased toxicity [96]. 

## 3. Subcellular Targets for Gene Delivery

CPPs have demonstrated remarkable ability to deliver diverse biomacromolecules into various plant species. The plasmid DNA delivery mediated by CPPs displayed a high potential and efficiency in plant root cells [67], embryos [97], and leaf cells [82] without protoplast preparations. Positively charged CPPs possess the abilities to interact with, condense, and package plasmid DNAs. The combination ratio between CPP and nucleic acid (Figure 2), also called nitrogen (NH_3_^+^)/phosphate (PO_4_^−^) (N/P) ratio [98], is key to DNA condensation and packaging. It further affects gene delivery efficiency [53,67]. An optimal N/P ratio makes CPP/DNA complexes more stable and is able to raise gene delivery efficiency. A good transgenic efficiency also depends on other factors, such as long-term stability of CPP/DNA complexes in cytosol, evasion from the endosome–lysosome system, targeted site of gene expression, and DNA releasing from CPP/DNA complexes [34,38,81,83,99]. The efficiency of cytoplasmic delivery by the predominant endosomal pathway is typically very low. A study showed that glutathione-responsive CPPs are able to escape from endosome entrapment and release DNAs at a higher rate to achieve gene transfer in plants [34]. Aside from efficiency, targeted delivery is also crucial in transgenic plant development [100]. Various DNA plasmids were designed and applied to the genes that were successfully achieved for development of transgenic plants (Table 2). Here, we discussed three major subcellular targets for CPP/DNA complex delivery: nucleus, plastids, and mitochondria (Figure 2).

### 3.1. Nucleus

Nuclear localization signal (NLS) is a small and basic peptide containing four lysines, one arginine, and several nonpolar residues [76] commonly found in CPP sequences (Table 1). Proteins or peptides containing this short signal are recognized by importin, and are transported into nucleus through the classical nuclear import pathway [76]. MPG, a chimeric CPP composed of HIV glycoprotein 41 and SV40 T antigen, is an example of NLS. NLS enters nucleus not only by itself but also with its cargoes [55]. According to the chemical and physical properties of NLS, it is considered a cationic peptide [27]. However, many studies suggested this NLS to be categorized as an amphipathic peptide, because its primary sequence contains both cationic and hydrophobic residues [22,55]. Fagerlund et al. suggested that lysine/arginine-rich NLS on signal transducers and activators of transcription 1 (STAT1) homodimer proteins and STAT1-STAT2 heterodimer proteins is key to both DNA binding and importin interaction [102]. Mutations of the conserved arginine/lysine-rich portions were able to prohibit nuclear import. Furthermore, R9-based CPPs (without NLS) affirmed the principal role of arginine in nuclear entry [66,69]. Both R9-green fluorescent protein (GFP) and NLS-R9-GFP displayed nuclear targeting in mung bean roots [67]. SR9 and PR9 entered cells via multiple pathways and classical endocytosis, respectively [21]. However, they all escaped from the endosome–lysosome system and entered nuclei [69]. Recent data studied by Kurnaeva et al. demonstrated that arginine residues are much more critical than lysines in NLS actions [103]. Therefore, NLS-tagged CPPs or arginine-rich CPPs play an important role in the nucleus delivery of nucleic acids, and their nuclear targeting abilities dramatically increase successful results in transgenic plants. 

### 3.2. Chloroplasts (Plastids)

Chloroplasts (a.k.a., plastids in plants) contain their own genomes and are the core components for photosynthesis. According to the membrane structure, plastids are divided into two groups: primary and secondary plastids [104]. Primary plastids are found in most algae and plants, while plankton typically belongs to the category of secondary plastid organisms. Plastid genomes are essential, as genes in plastids regulate not only metabolism of photosynthesis, but also energy transfer and storage [105]. Plastids also influence the expression of nuclear genes via plastid-to-nucleus signaling pathways, which regulate plastidic and extraplastidic processes to cope with environmental changes [105].

In recent years, transgenic plastids are gaining more attraction in biotechnology for the following reasons: (1) the genome in plastids is smaller than chromosomes in nuclei, only contains about 150 kb in molecular mass, and is easily manufactured by humans [106]; (2) a mature chloroplast contains a high copy number of circular double-stranded DNA, which is able to produce large amounts of recombinant proteins, which is very important for vaccine or economic production [107]; and (3) plastids are the maternal inheritance in most plant species. Plastid genetic engineering, such as in transplastomic plants, manipulates organellar DNA without changing the nuclear genes. Extranuclear genetic engineering prevents genetic pollution from the nucleus and protects wild-type plants or relative wild species [108]. 

Macromolecules tagged with a specific signal are essential for organelle-targeted delivery. Chloroplast transit peptides (CTPs, a.k.a. chloroplast-targeting peptides) are special peptides containing 33–35% hydrophobic, 22–23% hydroxylated, and 14–15% positively charged amino acids [109]. Shen et al. indicated that the most efficient CTPs in rice is RC2, and its sequence also follows the similar percentage of hydrophobic, hydroxylated, and cationic residues [77]. Thagun et al. successfully combined plasmid DNA, CTP (KH_9_-AtOEP34) [78], and CPP (BP100) [81] as a complex system to deliver DNAs into chloroplasts [99]. Further, they used the above complexes as nanocarriers, transfecting the plasmid DNA into chloroplasts after spraying on leaf surfaces [101]. The CTP/DNA complexes were transported from the extracellular space to the chloroplast stroma in *Arabidopsis* leaves [79]. These studies on the CPP(CTP)/DNA complex system dramatically enhanced the transgenesis without protoplast preparations nor callus formations, and provided a useful tool for rapid and effective plastids engineering in plants. 

### 3.3. Mitochondria

Mitochondrion is another valuable target in plant genetic modification. Mitochondria are the energy-producing organelles that contain the plasma membrane-like double membranes, their own genome (i.e., mitochondrial DNA; mtDNA), and a transcription-translation system [110]. mtDNA is a small and circular double-stranded DNA, similar to a plasmid DNA. This characteristic challenges scientists studying mtDNA modifications [110]. However, many factors, such as low transgenic efficiency, poor cytosolic entry, complicated preparation protocols, high mobility of mitochondria, and a limited number of cargo types, remain to be resolved before gene delivery into the mitochondria of plant cells can be widely used [28,83].

Foreign DNAs or cargoes tagged with mitochondrial targeting sequence (MTS) have a higher chance to be transported to mitochondria [28]. MTS, like CTP, is a short peptide signal, which is recognized by mitochondrial outer membrane receptor complex and interacts with components in mitochondrial protein import pathway [111]. The exciting mitochondrial transgenic results were first published by Chuah et al. [81]. They fused MTS with cationic lysine/histidine repeat residues ((KH)_9_), becoming the fused peptide MTP_KH_, and this MTP_KH_ formed complexes with CPPs (BP100). CPP/plasmid DNA complexes penetrated plasma membrane through CPPs, while MTP_KH_/DNA complexes were found to be localized into mitochondria [81]. The *Renilla* luciferase gene expression in mitochondria of *A. thaliana* illustrated that CPP-mediated gene delivery can be applied in mitochondrial transgenic engineering [81]. Recently, Xiao et al. developed two novel cell-penetrating mitochondrial-targeting Mito^Ligand^ ligands (miniCPM3 and SeSe-TPP) that contain 2~3 hydrophobic aromatic amino acids and 3~4 arginine residues [28]. This ligand design included the MTS conserved sequence, a hydrophobic-, and a cationic-rich amphipathic helix [111]. They found that Mito^Ligand^-delivered cargoes were predominantly localized inside mitochondria after cellular uptake and endosomal escape [28]. Artificial peptide (LURL)_3_ was another novel cell-penetrating MTS that demonstrates the importance of hydrophobicity and helicity for mitochondrial localization [83]. Together, these effective peptide-based methods provide a starting point for the development of more sophisticated plant mitochondrial transfection strategies.

## 4. Conclusions

Delivery and expression of exogenous genes in plants have economic potential in biotechnology and industry. Physical, chemical, and biological methods for gene deliveries have been developed for more than three decades. CPPs-based gene delivery systems bring a bright prospect for transgenesis in agriculture. In addition, this review sums up the classification of CPPs, cellular entry mechanisms, cytotoxicity, and various genes applied for development of transgenic plants. Varying designs on primary sequences of CPPs result in different cellular entry routes with different transfection efficiencies. CPPs can be modified for specific organelle-targeted delivery in plant cells. By targeting nuclei, chloroplasts, and mitochondria, CPPs/DNAs complexes elevate certain gene expressions of interest, which may increase higher agricultural yields.

## Figures and Tables

**Figure 1 molecules-28-03367-f001:**
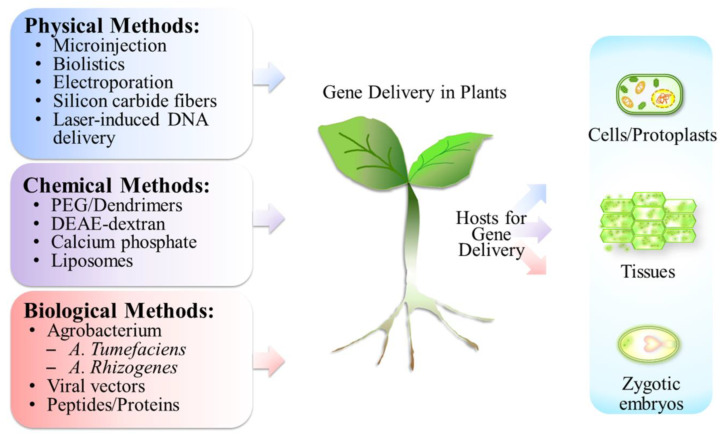
Methodologies of gene delivery in plants. Various methods including physical, chemical, and biological manners were applied in gene delivery. Plant cells were prepared as protoplasts for gene uptake. Plant tissues (callus) and zygotic embryos also served as the transgenic hosts. PEG: polyethylene glycol; DEAE: diethylaminoethanol.

**Figure 2 molecules-28-03367-f002:**
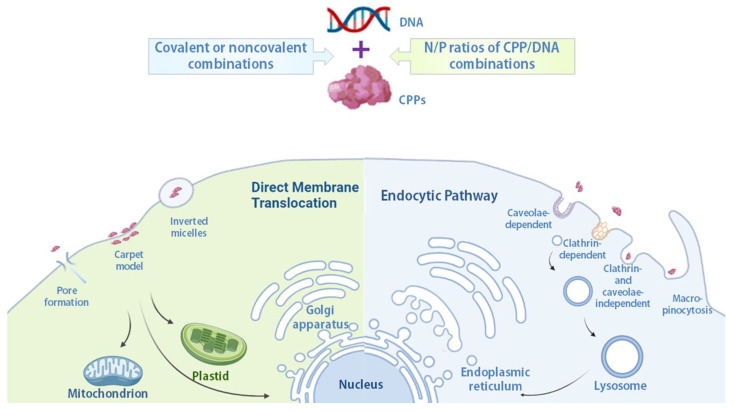
Cellular internalization mechanisms of CPP/DNA complexes and their subcellular targets in plant cells. There are two main routes of cellular internalization: endocytosis and direct membrane translocation. The internalized CPP/DNA complexes stay in the nucleus, plastids, or mitochondria, depending on signal sequences on CPPs. With endocytic pathway, CPP/DNA complexes have to escape from lysosomes eventually.

**Figure 3 molecules-28-03367-f003:**
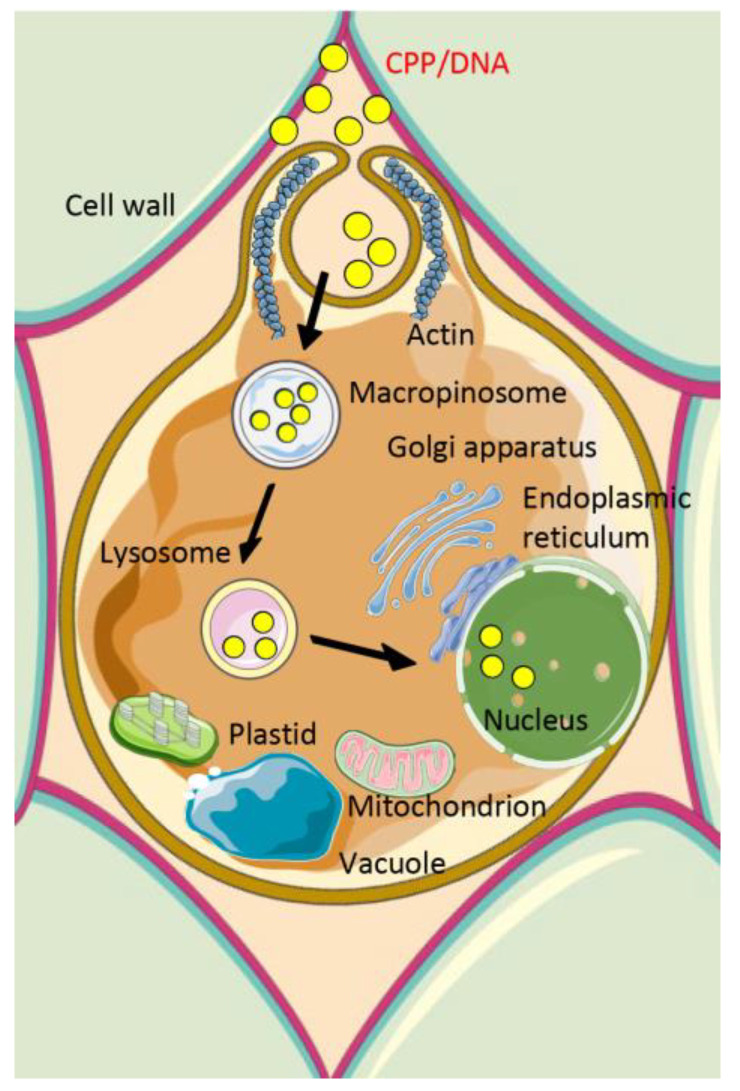
Schematic diagram of macropinocytosis.

**Table 1 molecules-28-03367-t001:** Comprehensive information of major cell-penetrating peptides and others.

CPP/Others	Primary Sequence	Categories	Target Cells/Tissues/Species	Entry Mechanisms	Cytotoxicity	References
L5a	RRWQW	Amphipathic	A549 cells	Direct membrane translocation	Up to 10 μM is not toxic	[53]
Tat PTD (48–60)	GRKKRRQRRRPPQ	Cationic	HeLa, HL116, CCL39 cells	Direct membrane translocation	Up to 100 μM is not toxic	[56]
			HeLa, CHO cells	Direct membrane translocation (pore formation)	1 μM, 3-day treatment without cytotoxicity	[57,58]
R9	RRRRRRRRR *	Cationic	Jurkat, murine B, human PBL cells	Endocytosis	–	[59]
			Plant tissues	Energy-independent pathway (link GFP with covalent manner)	Up to 2 μg is not toxic	[32]
pAntp	RQIKIWFQNRRMKWKK	Cationic	K562, HeLa cells	Direct membrane translocation (inverted micelle)	–	[60,61]
			CHO cells	Endocytosis (macropinocytosis)	Up to 10 μM is not toxic	[62]
VP22	DAATATRGRSAASRPTERPRAPARSASRPRRPVD	Amphipathic	CHO-K1, HeLa cells	Endocytosis (macropinocytosis)	–	[29,63]
C105Y	CSIPPEVKFNKPFVYLI	Hydrophobic	HuH7 cells	Energy-independent pathway (cell entry)	–	[55,64]
				Energy-dependent pathway (nucleolar entry)		
MAP	KLALKLALKALKAALKLA	Amphipathic	HeLa, endothelial cells	Energy-dependent pathway Energy-independent pathway	Up to 10 μM is not toxic –	[29,62,65]
HR9	CHHHHHRRRRRRRRRHHHHHC	Cationic	A549, Sf9, plant cells, paramecia, rotifers, prokaryotes	Direct membrane translocation	Up to 60 μM is not toxic	[21,66]
SR9	RRRRRRRRR	Cationic	A549 cells	– (Link protein with noncovalent manners)	Up to 16 μM is not toxic	[21]
			A549 cells	Multiple energy-dependent pathways (Link nanoparticles with noncovalent manners)	Up to 60 μM is not toxic	[26]
			Plant tissues	Macropinocytosis (link GFP or DNA with noncovalent manners)	Up to 16.6 μM is not toxic	[32,67]
			Prokaryotes	Macropinocytosis (Link nanoparticles with noncovalent manners)	Up to 48 μM is not toxic	[68]
PR9	FFLIPKGRRRRRRRRR	Cationic	A549 cells	Endocytosis (Link nanoparticles with noncovalent manners)	Up to 60 μM is not toxic	[69]
pVEC	LLIILRRRIRKQAHAHSK	Amphipathic	HeLa cells	Endocytosis	Up to 10 μM is not toxic	[62]
			Green alga	Direct membrane translocation	No toxicity	[70]
MPG	GALFLGFLGAAGSTMGAWSQPKKKRKV	Amphipathic	HS-68, Cos-7, HeLa cells	Direct membrane translocation	–	[29,71]
KALA	WEAKLAKALAKALAKHLAKALAKALKACEA	Amphipathic	CV-1, Hep G2, C2C12, K562, CaCo2 cells	Endocytosis	Toxic at the concentration ≥ 25 μM	[29,72]
GALA	WEAALAEALAAEALAEHLAEALAEALEALAA	Amphipathic	CV-1, Hep G2, C2C12, K562, CaCo2 cells	Endocytosis	–	[29,72]
Pep-1	KETWWETWWTEWSQPKKKRKV	Amphipathic	HeLa cells	Direct membrane translocation	–	[73,74]
NLS	CGYGPKKKRKVGG	Cationic (or amphipathic)	MCF-7, KB, HT29, MIAPACA2, PC3 cells	Energy-independent pathway	–	[75,76]
RC2	MQVWPIEGIKKFETLSYLPPL	Chloroplast transit peptide (CTP; not a CPP)	Rice chloroplasts	–	–	[77]
KH_9_-AtOEP34	KHKHKHKHKHKHKHKHKHMFAFQYLLVM	Cationic CPP combined with CTP	Seedlings and leaves of *A. thaliana* and *Nicotiana tabacum*	Endocytosis or direct membrane translocation (for cellular entry) Unknown (for plastid targeting)	–	[78,79]
BP100	KKLFKKILKYL	Amphipathic	Leaves of *A. thaliana*, BY-2 cells	Endocytosis	–	[35,80,81]
MTP-KH_9_	MLSLRQSIRFFKKHKHKHKHKHKHKHKHKH	Cationic CPP combined with MTS	Leaves of *A. thaliana*	Endocytosis (for cellular entry) Unknown (for mitochondrial targeting)	–	[81,82]
(LURL)_3_	LURLLURLLURL	Amphipathic MTS	Onion bulbs	*Gold nanoparticle biolistics* (for cellular entry) Binding to mitochondrial import receptors Tom20 and Tom22 (for mitochondrial targeting)	Negligible toxicity	[83]

* A segment of an *E. coli*-expressed fusion protein, while SR9 is a synthetic peptide.

**Table 2 molecules-28-03367-t002:** List of various genes applied in transgenic plants.

Delivery Methods	Genes	Targets	References
Non-CPP-based gene delivery	Proteinase inhibitor genes	Tobacco	[3]
	Recombinant Bt toxic proteins	*Vigna ungiguiculata*	[4,5]
	α-Amylase inhibitors, plant lectins	Adzuki bean	[5]
	Antibiotic-resistant Ti plasmid (*A. tumefaciens* mediated transfection)	Tobacco	[7]
	Hygromycin resistance gene	Protoplasts of rice	[14]
CPP-based gene delivery	p35S-RLuc-tNOS and p35S-GFP-tNOS plasmids	Leaves of *A. thaliana*	[34]
	p35S-Nluc-tNOS or p35S-GFP-tNOS plasmid	Seedlings of *A. thaliana*	[38]
	pHBT-sGFP(S65T)-NOS plasmid	Roots of mung bean and soybean	[67]
	psbAp:GFP:SPECr:psbAt At plastid genome integration vector, cox2p: GFP:SPECr:cox2t At mitochondrial genome integration vector, and cox2t:SPECr:GFP:cox2p Nt mitochondrial genome integration vector	Seedlings and leaves of *A. thaliana* or *N. tabacum*	[78]
	PsbA-SPECr-sGFP-psbA, Prrn-aadA-sfGFP-Trps, PsbA-SPECr-sGFP-psbA, and Prrn-aadA-sfGFP-Trps	Leaves of *A. thaliana*	[79]
	pDONR-cox2:rluc and pDONR-cox2:gfp plasmids	Leaves of *A. thaliana*	[81]
	pAct-1GUS plasmid	Wheat immature embryos	[97]
	pPrrn::GFP(S65T)::TpsbA, pPrrn::DsRed::TpsbA, and pPpsbA::Rluc plasmids	Leaves of *A. thaliana*	[99]
	psfGN155-MxMT and psfGC155-MxMT plasmids	Leaves of *N. benthamiana*	[100]
	pBI221, pBI121, and pPpsbA::Rluc plasmids	Leaves of *Arabidopsis*, soybean, and tomato	[101]

## Data Availability

Data available upon request from the corresponding author.

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
