# Peer review of "Cell-Penetrating Peptides for Use in Development of Transgenic Plants"

_molecules, 2023, doi:10.3390/molecules28083367_

Round 1

Reviewer 1 Report

The review article entitled “Cell-Penetrating Peptides for Use in Development of Trans- 2 genic Plants” addressing the nice contents, Figs. resolution is very high however English used in the text is very poor and need to revised thoroughly.

Few examples are as under,

1-      In introduction section paragraph on line 39-41 Modern agriculture is facing major……..may be revised and concise as the word “and” has been used about 5 times in same paragraph

2-      Various references are too old i.e., references from 19th that may be updated

3-      While writing Scientific names of plant species once may be write the complete name then abbreviations but the authors have written full names repeatedly in tables as well as in text i.e., ‘’Arabidopsis thaliana’’.

4-      Similar writing pattern has been used for microorganism’s names such as ‘’Agrobacterium tumefaciens’’ on line number 20, 53, 114 that need to be revised and check throughout the manuscript.

5-      List of genes may be added that indicating the genes that were successfully achieved for Development of Trans- 2 genic Plants.

6-      Conclusion section may be improved by mentioning key findings of this review.

Author Response

We appreciate your critical review and have carefully revised the manuscript according to your comments. Our revised manuscript has been significantly edited by a native English-speaking colleague. Your major comments are answered below:

  1. We have rephrased this sentence according to your suggestions in a more clear manner.
  2. Yes, we have updated both 18th and 19th
  3. Accordingly, we have checked and corrected scientific names of plant species in text and tables.
  4. As stated above, we have corrected these names throughout the entire manuscript.
  5. We have followed your comments to add a new Table 2 illustrating the genes used in transgenic plants.
  6. As suggested, we have re-written the Conclusion section in a more conclusive manner to include key findings of this review.

A detailed list of changes in the revised manuscript marked in red color is summarized below.

A list of changes:

            Line

Page*  number           Original version       Revised version

1          7          College of Acience and              College of Science and

1          17        with improved gains in yield…       with improved yield…

1          18        biotechnology that introduces…              biotechnology introducing…

1          32        In sum, the…                                   In summary, the…

1          38        Crop yield trends…                         The current trends in…

1          40        and plant pests...                              plant pests...

1          41        and all of ...                                     All of ...

2          52        in 1983 with an...                             in 1983 when an...

2          53        Agrobacterium tumefaciens             A. tumefaciens

2          59        plants in either a ...                          plants either directly ...

2          60        Another way of ...              Additional gene delivery...

2          68        tissue damages and                          tissue damage and

2          98        embryos were also served              embryos also served

3          104      The pros of ...                                  The benefits of …

3          109      ions with electrostatics,                   ions via electrostatics,

3          112      reduce the zeal of                            reduce the potential of

3          114      Agrobacterium tumefaciens             A. tumefaciens

3          116      deliver genes into host…                 deliver genes into the host…

3          121      Besides, it took long              Additionally, it took long

3          126      non-viral but peptide-based gene    non-viral peptide-based gene...

3          130      to detailly introduce...                      to introduce...in detail.

3          133      Cell membrane is a                          The cell membrane is a

3          138      nucleic acids into                             nucleic acids, into

3          142      improved while CPPs              improved when CPPs

3          148      Till now, studies                              Until now, studies

4          163      It further offered...              Additionally, it offered...

7          210      These are the pros of...                    As shown, there are...

7          211      However, the cons come              Unfortunately, the cons come

7          216      understanding about entry              understanding their entry

8          255      membrane with electrostatics              membrane utilizing electrostatics

8          263      while change the membrane...         while changing the membrane...

8          280      [22,63]. However, SR9                   [22,63], however, SR9

9          308      either leads to thin the                     either thins the

9          328      Kauffman et al. suggested               The authors also suggested

9          340      However, a good transgenic            A good transgenic

9          347      [100]. Here, we...                             [100]. Various DNA...

9          355      It enters nucleus not                        NLS enters nucleus not

10        372      3.2. Chloroplasts...                           Table 2...

10        384      and is easy to be manufactured        and is easily manufactured

10        396      sequence is also following the              sequence also follows the

10        399      and transfected the plasmid              transfecting the plasmid

11        411      This characteristic attack...              This characteristic attacks...

11        418      which is able to be recognized        which is recognized

11        425      of Arabidopsis thaliana                   of A. thaliana

11        440      agriculture. Varying...              agriculture. In addition, ...

11        442      By targeting to nuclei                      By targeting nuclei

11        455      Acknowledgments: None.              Acknowledgments: We...

12        486      Agrobacterium- and...              Agrobacterium- and...

12        496      18. Potrykus, I.                                18. Davey, M.R.; ...

12        497      19. Finer, K.R.; ...                            19. Hwang, H.H.; ...

13        550      In vivo proof of ...                           In vivo proof of ...

14        600      on Chlamydomonas                        on Chlamydomonas ...

14        623      of Arabidopsis thaliana                    of Arabidopsis thaliana

15        644      In silico approaches                         In silico approaches

 *Page and line numbers are referred to the original version.

Reviewer 2 Report

The article "Cell-Penetrating Peptides in Plant Delivery Systems: State of the Art and Future Directions" provides a comprehensive review of the use of cell-penetrating peptides (CPPs) in plant gene delivery systems. The authors discuss the current state of CPP-based gene delivery systems and their potential for use in agriculture. They describe the different types of CPPs and their cellular entry mechanisms, as well as modifications that can be made to CPPs for specific organelle-targeted delivery in plant cells. The authors also provide a brief overview of other gene delivery methods used in plants. Overall, this article provides valuable information for researchers interested in developing gene delivery systems using CPPs for plant biotechnology. Based on the detailed and thorough analysis presented in this article, I would like to recommend its acceptance for publication.

Minor concern:

Line 129: “We devote the sections below to detailly introduce CPPs and their roles in transgenic plants” to “We devote the sections below to introduce CPPs and their roles in transgenic plants in detail.”

Author Response

Thank you for the detailed review. We have carefully revised the original manuscript according to your valuable comments. Our answer to your minor concern is listed below:

  1. Line 129: We have rephrased this sentence according to your suggestion.

A detailed list of changes in the revised manuscript marked in red color is summarized below.

A list of changes:

            Line

Page*  number           Original version       Revised version

1          7          College of Acience and              College of Science and

1          17        with improved gains in yield…       with improved yield…

1          18        biotechnology that introduces…              biotechnology introducing…

1          32        In sum, the…                                   In summary, the…

1          38        Crop yield trends…                         The current trends in…

1          40        and plant pests...                              plant pests...

1          41        and all of ...                                     All of ...

2          52        in 1983 with an...                             in 1983 when an...

2          53        Agrobacterium tumefaciens             A. tumefaciens

2          59        plants in either a ...                          plants either directly ...

2          60        Another way of ...              Additional gene delivery...

2          68        tissue damages and                          tissue damage and

2          98        embryos were also served              embryos also served

3          104      The pros of ...                                  The benefits of …

3          109      ions with electrostatics,                   ions via electrostatics,

3          112      reduce the zeal of                            reduce the potential of

3          114      Agrobacterium tumefaciens             A. tumefaciens

3          116      deliver genes into host…                 deliver genes into the host…

3          121      Besides, it took long              Additionally, it took long

3          126      non-viral but peptide-based gene    non-viral peptide-based gene...

3          130      to detailly introduce...                      to introduce...in detail.

3          133      Cell membrane is a                          The cell membrane is a

3          138      nucleic acids into                             nucleic acids, into

3          142      improved while CPPs              improved when CPPs

3          148      Till now, studies                              Until now, studies

4          163      It further offered...              Additionally, it offered...

7          210      These are the pros of...                    As shown, there are...

7          211      However, the cons come              Unfortunately, the cons come

7          216      understanding about entry              understanding their entry

8          255      membrane with electrostatics              membrane utilizing electrostatics

8          263      while change the membrane...         while changing the membrane...

8          280      [22,63]. However, SR9                   [22,63], however, SR9

9          308      either leads to thin the                     either thins the

9          328      Kauffman et al. suggested               The authors also suggested

9          340      However, a good transgenic            A good transgenic

9          347      [100]. Here, we...                             [100]. Various DNA...

9          355      It enters nucleus not                        NLS enters nucleus not

10        372      3.2. Chloroplasts...                           Table 2...

10        384      and is easy to be manufactured        and is easily manufactured

10        396      sequence is also following the              sequence also follows the

10        399      and transfected the plasmid              transfecting the plasmid

11        411      This characteristic attack...              This characteristic attacks...

11        418      which is able to be recognized        which is recognized

11        425      of Arabidopsis thaliana                   of A. thaliana

11        440      agriculture. Varying...              agriculture. In addition, ...

11        442      By targeting to nuclei                      By targeting nuclei

11        455      Acknowledgments: None.              Acknowledgments: We...

12        486      Agrobacterium- and...              Agrobacterium- and...

12        496      18. Potrykus, I.                                18. Davey, M.R.; ...

12        497      19. Finer, K.R.; ...                            19. Hwang, H.H.; ...

13        550      In vivo proof of ...                           In vivo proof of ...

14        600      on Chlamydomonas                        on Chlamydomonas ...

14        623      of Arabidopsis thaliana                    of Arabidopsis thaliana

15        644      In silico approaches                         In silico approaches

 *Page and line numbers are referred to the original version.

Round 2

Reviewer 1 Report

Authors have incorporated the all the mentioned garamatical/technical errors and revised version is fine to accept